# Influence of Chitosan, Salicylic Acid and Jasmonic Acid on Phenylpropanoid Accumulation in Germinated Buckwheat (*Fagopyrum esculentum* Moench)

**DOI:** 10.3390/foods8050153

**Published:** 2019-05-06

**Authors:** Chang Ha Park, Hyeon Ji Yeo, Ye Eun Park, Se Won Chun, Yong Suk Chung, Sook Young Lee, Sang Un Park

**Affiliations:** 1Department of Crop Science, Chungnam National University, 99 Daehak-ro, Yuseong-gu, Daejeon 34134, Korea; parkch804@gmail.com (C.H.P.); guswl7627@gmail.com (H.J.Y.); yeney1996@cnu.ac.kr (Y.E.P.); seaw613@cnu.ac.kr (S.W.C.); 2Department of Plant Resource and Environment, Jeju National University, 102 Jejudaehak-ro, Jeju-si, Jeju Special Self-Governing Province 63243, Korea; yschung@jejunu.ac.kr; 3Marine Bio Research Center, Chosun University, 61-220 Myeongsasimni, Sinji-myeon, Wando-gun, Jeollanamdo 59146, Korea

**Keywords:** germinated buckwheat, elicitors, phenolic compounds, jasmonic acid, chitosan

## Abstract

The present study investigated the effects of jasmonic acid (JA), chitosan, and salicylic acid (SA) on the accumulation of phenolic compounds in germinated buckwheat. A total of six phenolics were detected in the buckwheat treated with different concentrations of SA (50, 100, and 150 mg/L), JA (50, 100, and 150 μM), and chitosan (0.01, 0.1, and 0.5%) using high-performance liquid chromatography (HPLC). The treatment with 0.1% chitosan resulted in an accumulation of the highest levels of phenolic compounds as compared with the control and the 0.01 and 0.5% chitosan treatments. The treatment with 150 μM JA enhanced the levels of phenolics in buckwheat sprouts as compared with those observed in the control and the 50 and 100 μM JA-treated sprouts. However, the SA treatment did not affect the production of phenolic compounds. After optimizing the treatment concentrations of elicitors (chitosan and JA), a time-course analysis of the phenolic compounds detected in the germinated buckwheat treated with 0.1% chitosan and 150 μM JA was performed. Buckwheat treated with 0.1% chitosan for 72 h showed higher levels of phenolic compounds than all control samples. Similarly, the germinated buckwheat treated with JA for 48 and 72 h produced higher amounts of phenolic compounds than all control samples. This study elucidates the influence of SA, JA, and chitosan on the production of phenolic compounds and suggests that the treatment with optimal concentrations of chitosan and JA for an optimal time period improved the production of phenolic compounds in germinated buckwheat.

## 1. Introduction

*Fagopyrum esculentum* Moench (common buckwheat), belonging to the Polygonaceae family, is an important pseudocereal cultivated and consumed in East Asian countries. It has high agricultural and medicinal values [1]. It contains various minerals (magnesium, copper, zinc, and manganese), fiber, and a large quantity of rutin [2], which exhibits anti-allergic [3], cytoprotective [4], anti-thrombotic [5], and anti-carcinogenic activities [6]. Furthermore, rutin and its related flavonoids in buckwheat have various health effects. For example, it functions as an inhibitor of cardiovascular problems, such as arteriosclerosis disease, high blood pressure, and capillary fragility [2].

Dietary fibers and phenolics are plant food constituents that play a beneficial role in human health, and use of these constituents as functional ingredients has gradually increased [7]. These constituents are usually studied separately due to differences in their metabolic pathways, physicochemical and biological properties, and chemical structures [8]. Recent studies, however, have reported that phenolics, as fiber copassengers, are bound to the fiber fraction and can be released along the gastrointestinal (GI) tract [9,10]. In particular, cereal dietary fibers with phenolics may play a role in antioxidant protection at the intestinal environmental level. In the GI tract, free phenolics are generally released from soluble dietary fibers by the activities of microbial and intestinal enzymes, such as esterases, and then absorbed through the intestine. Such a continuous absorption of phenolics can explain that the high consumption of whole grain can reduce the risk for developing diabetes, cancer, and cardiovascular diseases [9,10,11]. 

Flavonoids are well-known polyphenolic compounds consisting of a benzo-γ-pyrone structure and are commonly found in plant species. They are derived from the phenylpropanoid pathway [12]. These phenolic compounds are usually distributed in plant parts, including roots, stems, leafs, flowers, and fruits, herbs, vegetables, and nuts. These secondary metabolites are well-known components of food sources used in the daily human diet [13]. They exhibit various health benefits such as anti-inflammatory [14], antitumor, anti-human immunodeficiency virus [15], anti-tuberculosis [16], and anti-diabetic activities [17].

The accumulation of secondary metabolites is activated by abiotic stresses, signal molecules, or elicitors in various plants [18]. In particular, the production of secondary metabolites can be promoted by the elicitations by chitosan, salicylic acid, and jasmonic acid and by the ultraviolet-A/B radiation [19]. Chitosan elicitation leads to an increase in the production of phenylpropanoids. In chitosan-elicitated cells of *Cocos nucifera* (coconut), the production of phenolic compounds was enhanced in the cell suspension cultures [20]. Likewise, salicylic acid (2-hydroxybenzoic acid) from intact grape berries and jasmonic acid from the cells of *Hypericum perforatum* L. (St. John’s wort) led to an increase in the total phenolic content. In particular, a rapid increase in the concentration of phenolic compounds was observed in JA-elicited cells compared to the control cells after 4 days of jasmonic acid (JA) elicitation [21]. An irradiation treatment with ultraviolet-A (UV-A) activated phenylalanine ammonia lyase (PAL), a key enzyme in the phenylpropanoid biosynthetic pathway, in tomato seedlings in addition to anthocyanin production in hypocotyls and fruit [22]. Furthermore, increase in hypericin and hyperforin accumulation was observed in *H. perforatum* exposed to ultraviolet-B (UV-B) radiation [23]. 

To our knowledge, no previous reports have documented the influence of chitosan, salicylic acid, and jasmonic acid on the accumulation of flavonoids in germinated buckwheat. Thus, the current study aimed to elucidate the effect of chitosan, salicylic acid, and jasmonic acid on the production of phenolic compounds in germinated buckwheat. 

## 2. Materials and Methods

### 2.1. Plant Materials

Seeds of common buckwheat were obtained from Jeju Buckwheat Farmers Association Corp. (Je-ju do, Korea). Two hundred seeds (approximately 4 g) were placed on filter paper (Whatman, 150-mm diameter) in a Petri dish (AccuResearch Korea, Seoul, South Korea, 150 mm diameter) and then treated with 200 mL of salicylic acid at concentrations of 50, 100, and 150 mg/L, jasmonic acid at concentrations of 50, 100, and 150 μM, and chitosan at concentrations of 0.01, 0.1, and 0.5% (Figure 1). After incubation under a dark condition at 25 °C for 72 h, the germinated buckwheat seeds were harvested and frozen in liquid nitrogen (−196 °C). Individual samples were lyophilized and finely ground for further analysis. All samples were prepared in triplicate.

### 2.2. Extraction and High-Performance Liquid Chromatography Analysis of Phenolics

The extraction and high-performance liquid chromatography (HPLC) analysis of phenolics in germinated buckwheat were performed according to the slightly modified method described by Park et al. [1]. For the extraction of phenolic compounds, 0.2 g of individual sample was soaked in 2 mL of aqueous methanol (80% *v*/*v*) and vortexed for 30 s. After sonication in a water bath at 37 °C for 60 min, the sample was centrifuged at 16,000× *g* for 15 min, and the first supernatant was obtained. Additionally, the entire procedure was carried out twice. The collected supernatants were evaporated and then resuspended in 2 mL of methanol. The extracts were passed thorough a 0.45 µm syringe filter into an HPLC vial. The analytical equipment and conditions for the HPLC analysis were performed as described by Park et al. [1]. The phenolic compounds were identified based on the retention time and spike test, followed by a calculation using respective calibration curves. The linear equations were *y* = 7.5252*x* − 37.3870 (*R*^2^ = 0.9997, recovery value = 102.81 ± 5.32%) for benzoic acid, *y* = 39.9829*x* − 65.7075 (*R*^2^ = 0.9999, recovery value = 102.14 ± 3.67%) for caffeic acid, *y* = 7.8897*x* − 40.2424 (*R*^2^ = 0.9999, recovery value = 104.09 ± 11.25%) for catechin, *y* = 8.5989*x* − 8.3356 (*R*^2^ = 0.9999, recovery value = 100.28 ± 0.80%) for epi-catechin, *y* = 32.8959*x* − 26.1737 (*R*^2^ = 0.9999, recovery value = 96.57 ± 2.51%) for gallic acid, and *y* = 8.0971*x* − 105.5466 (*R*^2^ = 0.9995, recovery value = 104.61 ± 11.17%) for rutin. The chemical structures of the compounds are shown in Appendix A. The external standards were purchased from Sigma-Aldrich Co., Ltd. (St. Louis, MO, USA). The results were presented as microgram per milligram dry weight (μg/mg (dw)) with means ± standard deviation of triplicate experiments. 

### 2.3. Statistical Analysis

Analysis of variance (ANOVA) test evaluates the statistical data and Duncan’s multiple range test (DMRT) at *p* < 0.05 were carried out using the SAS software (version 9.4, 2013; SAS Institute, Inc., Cary, NC, USA).

## 3. Results

### 3.1. Effects of Elicitor Treatments on Germinated Buckwheat

Table 1 shows the effect of elicitors used during germination on the accumulation of phenolic compounds in buckwheat sprouts. Six phenolic compounds (caffeic acid, catechine, chlorogenic acid, (−)-epicatechine, gallic acid, and rutin) and one organic acid (benzoic acid) were detected in germinated buckwheat. Even though benzoic acid does not belong to phenolic compound, the total phenolic compound of all samples were described, including benzoic acid. The treatment with 0.1% chitosan increased the total phenolic content compared with the control and the 0.01 and 0.5% chitosan treatments (*p* < 0.05). In particular, the total phenolics of the germinated buckwheat treated with 0.1% chitosan were approximately 1.23 times higher than the control buckwheat samples (Table 1). In addition, the concentration of gallic acid, catechin, chlorogenic acid, and (−)-epicatechin in the germinated buckwheat treated with 0.1% chitosan were approximately 15.86, 1.72, 1.64, and 2.17 times higher than those of the control. 

The six phenolic compounds were also detected by HPLC in buckwheat treated by JA. The germinated buckwheat treated with JA at the specific concentrations of 50, 100, and 150 µM increased the accumulation of total phenolic compounds. The germinated buckwheat grown in 150 µM of JA showed the highest amount of total phenolics which was approximately 2.47 times higher than that of control. Particularly, the accumulation of gallic acid, rutin, catechin, chlorogenic acid, and (−)-epicatechin were approximately 2.00, 2.38, 1.76, 2.81, and 7.95 times higher in JA-treated buckwheat than in the control buckwheat samples (Table 1). However, SA treatment did not influence the production of phenolic compounds.

### 3.2. Time-Course Effects of 0.1% Chitosan Treatment on Phenolic Compounds of Germinated Buckwheat

Owing to the highest accumulation of total phenolics after 0.1% chitosan treatment, the accumulation of phenolic compounds in 0.1% chitosan-treated buckwheat was studied throughout the germination process (6, 12, 24, 48, and 72 h). As a result, the concentration of total phenolic compounds increased after 72 h, and it was the highest accumulation of phenolics compared with all the control groups (6, 12, 24, 48, and 72 h). In particular, at 72 h, the amounts of gallic acid, chlorogenic acid, (−)-epicatechin, and rutin were higher in the chitosan-treated buckwheat than in the buckwheat under control (Table 2). However, the fresh and dry weight (g) of germinated buckwheats after 72 h were not significantly different compared with those of the control (Appendix A).

### 3.3. Time-Course Effects of 150 µM Jasmonic Acid Treatment on Phenolic Compounds of Germinated Buckwheat

The time course experiments at 6, 12, 24, 48, and 72 h were also conducted in buckwheat germinated in the presence of 150 μM jasmonic acid. As a result, the accumulation of total phenolic compounds increased after 72 h. Particularly, the germinated buckwheat treated with 150 µM jasmonic acid for 72 h showed the highest concentration of total phenolics compared with the control buckwheat samples. Likewise, gallic acid, chlorogenic acid, (−)-epicatechin, and rutin showed the highest levels in the jasmonic acid-treated germinated buckwheat as compared with the control and other treatments (Table 2). However, the fresh and dry weight (g) of germinated buckwheats after 72 h were not significantly different compared with those of the control (Appendix A).

## 4. Discussion

In this study, six phenolic compounds (gallic acid, catechin, chlorogenic acid, caffeic acid, (−)-epicatachin, and rutin) and one organic acid (benzoic acid) were detected in germinated buckwheat. These results are consistent with previous studies reporting the identification of gallic acid, chlorogenic acid, catechin, caffeic acid, (−)-epicatechin, and rutin in common buckwheat sprouts [24] and flours [25]. Furthermore, the presence of caffeic and benzoic acids was recorded in buckwheat honeys and four different phenolics, including catechin, chlorogenic acid, epicatechin, and rutin, were identified in different parts, such as stem, leaf, flower, and root, of the Korean common buckwheat cultivars [26]. 

This time course analysis revealed that chitosan and JA gradually enhanced the production of phenolic compounds in the germinated buckwheat. We carefully suggested that it might be due to increased gene expression levels of phenylpropanoid-related genes by the chitosan and JA elicitation since our previous studies reported that the methyl jasmonate increased gene expression levels of phenlypropanoid-related genes and enhanced the accumulation of phenolic compounds in radish sprouts [27] and in *Agastache rugosa* Kuntze [28], respectively. Furthermore, Chen et al. (2009) reported the increased expression of phenylpropanoid and flavonoid biosynthesis genes and in soybean sprouts treated with chitosan [29]. 

Among the detected phenolics in the germinated buckwheat at 72 h after the treatment of 150 µM JA and 0.1% chitosan, the concentration of rutin, (−)-epicatechin, and chlorogenic acid significantly increased. Rutin, the most abundant phenolic compound in the elicited germinated buckwheat, is used as a health supplement and has applications in food industries due to its biological activities, including anti-oxidant, anti-inflammatory, and anti-diabetic activities [30]. Similarly, (−)-epicatechin, the second most abundant compound, has been introduced as a health supplement because it enhances fatigue resistance and oxidative capacity [31,32]. Chlorogenic acid, the third most abundant compound, has been mainly used in food processing and cosmetic industries since the compound possesses anti-carcinogenic [33], anti-inflammatory [34], and anti-oxidant functions [35]. The other identified compounds have been reported to have health-beneficial effect, such as anti-cancer and anti-oxidant effects [36,37,38].

Elicitation is considered one of the best strategies to stimulate secondary metabolites. The accumulation of secondary metabolites from either parts of parent or transformed plants is greatly dependent on the sources of their origin; however, it might be influenced by the treatments as well as environmental factors. Elicitors, when in contact with the cells of higher plants, trigger an increase in the production of pigments, flavones, phytoalexins, and other defense-related compounds [39,40,41,42]. This study revealed that treatment with elicitors chitosan or jasmonic acid can enhance the production of phenolic compounds in germinated buckwheat. This finding was consistent with previous studies of Park et al. (2017) [1] and Kim et al. (2011) [43], who reported the enhancement of phenolics in the sprouts of common buckwheat by treatment with indoleacetic acid and methyl jasmonic acid, respectively. Li et al. (2015) [44] reported the positive effect of the exogenous application of sucrose on the flavonoid contents of common buckwheat seedlings. Lim et al. (2012) [45] reported that the sodium chloride (NaCl) treatment enhanced both phenylpropanoid and carotenoid production in buckwheat sprouts. In addition, elicitors could stimulate the biosynthesis of phenylpropanoid compounds in tartary buckwheat (*F. tataricum* (L.) Gaertn.). Zhao et al. (2015) [46] reported an increase in flavonoid production in sprout cultures under treatment of polysaccharide elicitors. Sun et al. (2012) [47] reported that the treatment with salicylic acid resulted in an increase in rutin production in tartary buckwheat leaves. Li et al. (2017) [48] also described that exogenous ethephon application enhanced phenylpropanoid biosynthesis. Furthermore, Park et al. (2016) [49] reported that treatment with auxins improved anthocyanin production in the hairy root cultures of tartary buckwheat.

Chitosan and JA affect the phenlypropanoid biosynthesis in plants. Previous studies reported that chitosan increases the activity of key enzymes (phenylalanine ammonia-lyase (PAL) and tyrosine ammonia-lyase) of the phenylpropanoid pathway [50], and JA also increases PAL activity [51]. Furthermore, chitosan treatment enhanced the accumulation of free and bound phenolic acids in peanut seeds [52]. Mandal et al. (2016) [53] reported that chitosan increased the production of cell wall–bound phenolic compounds in *Solanum Melongena*. Moreover, methyl jasmonic acid could increase these phenolic compounds [53]. Additionally, Skrzypczak-Pietraszek et al. (2014) [54] reported that the production of free and bound phenolic acids increased by adding methyl jasmonic acid in shoot cultures of *Exacum affine* Balf. f. ex Regel. Therefore, we carefully suggest that JA and chitosan treatment enhance the production of free and bound phenolic compounds in plants.

## 5. Conclusions

This study confirmed that JA and chitosan play an important role in the production of phenolic compounds in germinated buckwheat. A total of six phenolics (gallic acid, catechin, chlorogenic acid, caffeic acid, (−)-epicatechin, and rutin) and one organic acid (benzoic acid) were detected in germinated buckwheat. JA and chitosan treatment enhanced the accumulation of phenolic compounds in the germinated buckwheat. Particularly, treatments with 150 µM JA were the most effective on the accumulation of phenolic compounds. According to the time-course analysis, a 72 h chitosan treatment enhanced the production of phenolics. Similarly, the germinated buckwheat treated for 48 and 72 h showed a higher accumulation of phenolic compounds than control buckwheat. Thus, these results might help build sturdy strategies for enhancing the production of phenolics in germinated buckwheat as a good nutritional source for human consumption. 

## Figures and Tables

**Figure 1 foods-08-00153-f001:**
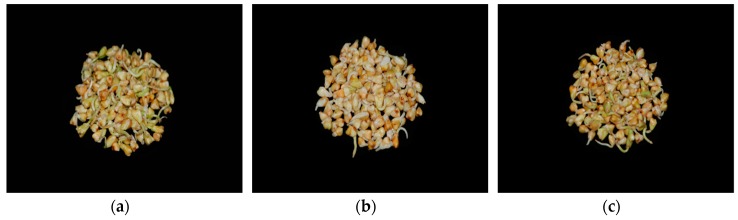
Buckwheat germinated for 72 h. (**a**) Control; (**b**) germinated buckwheat treated with 150 μM jasmonic acid; (**c**) germinated buckwheat treated with 0.1% chitosan.

**Table 1 foods-08-00153-t001:** The effect of elicitors (chitosan, jasmonic acid (JA), salicylic acid (SA)) on the accumulation of phenolic compounds (μg/g (dw)).

	Benzoic Acid	Caffeic Acid	Catechin	Chlorogenic Acid	(−)-Epicatechin	Gallic Acid	Rutin	Total
Control	74.48 ± 5.27 ^abc^^,^^1^	77.99 ± 1.54 ^cd^	56.18 ± 2.37 ^c^	58.92 ± 1.52 ^d^	44.44 ± 8.55 ^d^	6.09 ± 0.19 ^c^	424.42 ± 0.96 ^cde^	736.43 ± 11.35 ^e^
Chitosan 0.01%	71.34 ± 4.48 ^bc^	82.52 ± 8.84 ^cd^	64.32 ± 6.63 ^c^	81.62 ± 11.69 ^bc^	48.79 ± 27.38 ^d^	6.27 ± 0.44 ^c^	399.7 ± 53.28 ^def^	754.55 ± 105.14 ^e^
Chitosan 0.1%	58.17 ± 4.38 ^d^	81.25 ± 4.12 ^cd^	96.59 ± 8.06 ^b^	99.66 ± 2.91 ^b^	98.51 ± 17.90 ^c^	9.19 ± 1.91 ^b^	465.76 ± 50.35 ^cd^	909.12 ± 76.28 ^d^
Chitosan 0.5%	68.56 ± 9.03 ^bcd^	70.27 ± 5.52 ^d^	66.34 ± 9.52 ^c^	66.56 ± 3.16 ^cd^	24.96 ± 18.03 ^d^	5.61 ± 0.45 ^c^	341.12 ± 35.09 ^f^	643.43 ± 26.63 ^e^
JA 50 µM	59.49 ± 2.84 ^d^	96.61 ± 7.19 ^bc^	104.71 ± 10.35 ^b^	150.7 ± 23.68 ^a^	297.41 ± 53.66 ^b^	10.8 ± 2.00 ^ab^	494.99 ± 65.45 ^c^	1214.71 ± 153.05 ^c^
JA 100 µM	58.06 ± 6.47 ^d^	104.76 ± 17.15 ^ab^	136.12 ± 32.75 ^a^	155.34 ± 8.87 ^a^	299.5 ± 25.82 ^b^	10.5 ± 1.68 ^ab^	764.39 ± 39.19 ^b^	1528.66 ± 108.41 ^b^
JA 150 µM	68.17 ± 3.74 ^cd^	115.63 ± 11.79 ^a^	98.8 ± 23.94 ^b^	165.33 ± 22.43 ^a^	353.28 ± 13.17 ^a^	12.17 ± 0.85 ^a^	1011.3 ± 3.11 ^a^	1824.69 ± 72.80 ^a^
SA 50 mg/L	79.48 ± 10.83 ^ab^	65.19 ± 19.09 ^d^	61.94 ± 4.92 ^c^	56.08 ± 5.78 ^d^	31.55 ± 7.38 ^d^	6.83 ± 0.28 ^c^	375.63 ± 72.65 ^ef^	676.7 ± 111.09 ^e^
SA 100 mg/L	76.12 ± 2.94 ^abc^	73.11 ± 6.12 ^d^	58.92 ± 1.2 ^c^	58.98 ± 1.86 ^d^	37.62 ± 4.64 ^d^	6.24 ± 1.14 ^c^	420.9 ± 32.14 ^cdef^	731.89 ± 42.04 ^e^
SA 150 mg/L	84.75 ± 1.76 ^a^	78.63 ± 0.51 ^cd^	59.4 ± 1.17 ^c^	62.21 ± 1.71 ^cd^	44.93 ± 2.32 ^d^	7.24 ± 0.13 ^c^	456.62 ± 6.31 ^cde^	793.79 ± 7.02 ^de^

^1^ Means with different letters in the same column differ significantly (*p* < 0.05, Duncan multiple range test (DMRT)).

**Table 2 foods-08-00153-t002:** High-performance liquid chromatography (HPLC) analysis of total phenolic compounds in the germinated buckwheat under 150 μM jasmonic acid and 0.1% chitosan time-course treatment (μg/g (dw)).

	Benzoic Acid	Caffeic Acid	Catechin	Chlorogenic Acid	(−)-Epicatechin	Gallic Acid	Rutin	Total
Control 6 h	81.96 ± 4.91 ^abc,^^1^	74.93 ± 2.37 ^def^	57.21 ± 2.11 ^c^	55.72 ± 2.24 ^c^	43.13 ± 5.89 ^de^	6.78 ± 0.86 ^defgh^	371.84 ± 81.94 ^d^	691.57 ± 97.47 ^de^
Control 12 h	83.82 ± 3.10 ^ab^	74.29 ± 2.57 ^def^	66.30 ± 3.85 ^c^	59.91 ± 3.24 ^c^	39.60 ± 1.00 ^de^	8.03 ± 0.45 ^cd^	428.66 ± 6.33 ^cd^	760.60 ± 6.96 ^de^
Control 24 h	83.27 ± 1.96 ^ab^	76.44 ± 3.48 ^cdef^	65.97 ± 2.04 ^c^	61.52 ± 0.97 ^c^	34.10 ± 1.41 ^de^	6.98 ± 0.36 ^defgh^	418.00 ± 16.50 ^cd^	746.28 ± 23.56 ^de^
Control 48 h	71.64 ± 9.63 ^de^	84.56 ± 8.24 ^bc^	99.05 ± 15.37 ^b^	105.81 ± 32.67 ^b^	135.74 ± 95.97 ^c^	7.91 ± 1.56 ^cde^	291.20 ± 100.31 ^e^	795.91 ± 95.38 ^cd^
Control 72 h	74.48 ± 5.27 ^cde^	77.99 ± 1.54 ^cde^	56.18 ± 2.37 ^c^	58.92 ± 1.52 ^c^	44.44 ± 8.55 ^de^	6.09 ± 0.19 ^h^	424.42 ± 0.96 ^cd^	736.43 ± 11.35 ^de^
Chitosan 6 h	80.78 ± 2.42 ^abc^	69.16 ± 1.78 ^ef^	52.53 ± 0.87 ^c^	57.24 ± 1.58 ^c^	23.58 ± 2.47 ^e^	7.83 ± 0.63 ^cde^	346.41 ± 12.92 ^de^	637.53 ± 14.44 ^e^
Chitosan 12 h	80.59 ± 3.57 ^abc^	67.95 ± 1.60 ^f^	54.11 ± 3.32 ^c^	55.47 ± 0.79 ^c^	20.00 ± 3.43 ^e^	6.33 ± 0.22 ^fgh^	346.22 ± 23.57 ^de^	630.67 ± 34.24 ^e^
Chitosan 24 h	80.64 ± 0.57 ^abc^	67.89 ± 2.22 ^f^	57.30 ± 0.63 ^c^	55.36 ± 0.04 ^c^	21.26 ± 1.66 ^e^	6.45 ± 0.25 ^efgh^	367.70 ± 26.01 ^de^	656.61 ± 25.54 ^de^
Chitosan 48 h	78.79 ± 0.43 ^abcd^	73.47 ± 1.71 ^def^	70.25 ± 9.12 ^c^	72.28 ± 9.49 ^c^	55.14 ± 28.34 ^de^	6.14 ± 0.29 ^gh^	408.89 ± 15.52 ^cd^	764.96 ± 63.46 ^de^
Chitosan 72 h	58.17 ± 4.38 ^f^	81.25 ± 4.12 ^cd^	96.59 ± 8.06 ^b^	99.66 ± 2.91 ^b^	98.51 ± 17.90 ^cd^	9.19 ± 1.91 ^bc^	465.76 ± 50.35 ^c^	909.12 ± 76.28 ^c^
Jasmonic acid 6 h	77.14 ± 2.18 ^abcd^	76.42 ± 4.33 ^cdef^	55.06 ± 0.83 ^c^	60.78 ± 0.41 ^c^	39.81 ± 6.70 ^de^	7.58 ± 0.09 ^defg^	408.23 ± 16.91 ^cd^	725.03 ± 29.48 ^de^
Jasmonic acid 12 h	76.13 ± 2.83 ^bcd^	74.11 ± 1.01 ^def^	56.20 ± 0.54 ^c^	69.99 ± 1.64 ^c^	40.19 ± 0.48 ^de^	8.04 ± 0.26 ^cd^	418.48 ± 18.94 ^cd^	743.15 ± 20.99 ^de^
Jasmonic acid 24 h	74.63 ± 2.15 ^cde^	80.45 ± 2.57 ^cd^	62.54 ± 2.92 ^c^	70.34 ± 0.60 ^c^	38.88 ± 6.56 ^de^	7.69 ± 0.42 ^def^	395.77 ± 18.89 ^cd^	730.30 ± 22.81 ^de^
Jasmonic acid 48 h	84.85 ± 3.43 ^a^	91.14 ± 5.84 ^b^	165.89 ± 51.97 ^a^	109.62 ± 20.75 ^b^	202.38 ± 89.88 ^b^	9.79 ± 0.35 ^b^	541.76 ± 68.57 ^b^	1205.4 ± 240.79 ^b^
Jasmonic acid 72 h	68.17 ± 3.74 ^e^	115.63 ± 11.79 ^a^	98.80 ± 23.94 ^b^	165.33 ± 22.43 ^a^	353.8 ± 13.17 ^a^	12.17 ± 0.85 ^a^	1011.30 ± 3.11 ^a^	1824.6 ± 72.80 ^a^

^1^ Means with different letters in the same column differ significantly (*p* < 0.05, Duncan multiple range test (DMRT)).

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
