# Peer review of "Influence of Chitosan, Salicylic Acid and Jasmonic Acid on Phenylpropanoid Accumulation in Germinated Buckwheat (*Fagopyrum esculentum* Moench)"

_foods, 2019, doi:10.3390/foods8050153_

Round 1
Reviewer 1 Report
Manuscript foods-452364 “Influence of Chitosan and Jasmonic Acid on Phenylpropanoid Accumulation in Germinated Buckwheat (Fagopyrum esculentum Moench)” investigates the effects of elicitors, jasmonic acid (JA), chitosan, and salicylic acid (SA), on the accumulation of seven phenolic compounds in germinated buckwheat. The use of elicitors represents an important tool in order to modify the accumulation of secondary metabolites in various plants and thus represent an interesting subject.
Although there was some effort from the authors the manuscript lacks novelty and is not well written. There are issues related to the extraction method used to obtain phenolics as well as the validation of the HPLC method used. Moreover, a language check by a native speaker is highly recommended. During reviewing of the present manuscript are raised some concerns that need to be addressed.
More in detail:
Abstract:
- Line 22: The treatment with 0.1% chitosan gives the highest level of phenolics among the concentrations of chitosan applied in this study. The way reported here gives the impression that results in the highest concentration oh phenolics among all the treatments.
- Line 22: A language issue. Please check the sentence "A treated with......"
- Lines 27-30: Sentence repeats the findings reported previously in lines 21-24.
Introduction:
- Lines 68-70: Reformulate the sentence. Do you mean that the increase in the phenylpropanoids was confirmed by HPLC.
- Line 71: Sentence "...... compounds compared to the control cells." Citing literature is missing.
Materials and Methods:
- Line 86: were obtained ….from.
- Line 90-91: Report more in detail the conditions of germination (i.e. dark, light etc)
- Section 2.2. You should report the phenolics analysed. Moreover, there is needed to report in detail the standards used. Since you do quantitative analysis, the method used should be validated for the substrate analysed in the present study. If you perform it you should report the data.
- Line 98: The temperature used 60°C could affect negatively the amount of phenolics. Why did you choose this temperature? Did you check that extraction in this temperature is safe?
- Line 103: In the abstract was mentioned that was monitored the increase in the content of "phenolic compounds" whereas here it was showed that phenylpropanoid content was identified. Rephrase the sentence.
- Section 2.3. There are not mentioned the groups that are compared but only the test.
Results:
- Section 3.1: The results are not explained properly. There is not reported first the effect of elicitors on the total concentration of phenolics and then in each phenolic compound. This could bring to the reader more information and show differences among the phenolics studied.
- Line 116: There are reported four numbers for concentration increase and only three concentration (control, 0.01%, 0.05%).
- Table1: Usually letters used to show differences among the means are assigned from the lowest to the highest value. (a assigned to the lowest value and c to the highest among three values). Report properly the letters. Moreover, the same column begins with the control and finishes with SA150mg/L. You have to report that you compare groups of the same treatment within the same column.
- Section 3.2: Change the title of the section in order to report the time effect. In the form reported here is not showing the progress during germination time.
- Line 132-134: You report that "As a result, the concentration of total phenolic compounds increased after 72 h and it was the highest accumulation of phenolics compared with the control treatment. " Comparison was made with control 72h? You should clarify it. You should report the results of this statistical analysis on the table 2. That is comparisons between the control and the 0.1% chitosan by using numbers as superscript. The same is valid and for the table 3
Discussion
- Discussion is merely a literature review and does not discuss the findings reported in the section Results. Moreover, it has repetitions. It reports that according to the literature rutin is the most abundant phenolic compound found in germinated buckwheat but they do not link this fact with the data of the present study. Similarly, the second most abundant compound is epicatechin but there is not mentioned the fact that in the present study there is not found in the second place.
Conclusions:
Here is reported that the aim of the present study is to optimize the concentrations of elicitors and treatment period but this is not showed in the introduction (The aim in the introduction is not the same with the one reported here). Moreover, the authors do not perform any optimization technique such as response surface methodology in both factors studied (elecitors concentration, treatment period) in order to support the fact that they optimized the process. Here is not showed what are the unique findings of the present study.
Author Response
Manuscript foods-452364 “Influence of Chitosan and Jasmonic Acid on Phenylpropanoid Accumulation in Germinated Buckwheat (Fagopyrum esculentum Moench)” investigates the effects of elicitors, jasmonic acid (JA), chitosan, and salicylic acid (SA), on the accumulation of seven phenolic compounds in germinated buckwheat. The use of elicitors represents an important tool in order to modify the accumulation of secondary metabolites in various plants and thus represent an interesting subject.
Although there was some effort from the authors the manuscript lacks novelty and is not well written. There are issues related to the extraction method used to obtain phenolics as well as the validation of the HPLC method used. Moreover, a language check by a native speaker is highly recommended. During reviewing of the present manuscript are raised some concerns that need to be addressed.
Answer: Thank you for your valuable comment. We are honored to receive your feedback and we are sure that your comment will significantly improve the paper.
- Line 22: The treatment with 0.1% chitosan gives the highest level of phenolics among the concentrations of chitosan applied in this study. The way reported here gives the impression that results in the highest concentration oh phenolics among all the treatments.
Answer: Thank you for your help. We have changed this part. Now it reads “The treatment with 0.1 % chitosan resulted in the accumulation of the highest levels of phenolic compounds as compared with the control- and 0.01 and 0.5 % chitosan treatments.”
- Line 22: A language issue. Please check the sentence "A treated with......"
Answer: Thank you for your help. We have corrected this part. Now it reads “The treatment with 150 μM JA enhanced the levels of phenolics in buckwheat sprouts as compared with those observed in control, 50 and 100 μM JA-treated sprouts.”
- Lines 27-30: Sentence repeats the findings reported previously in lines 21-24.
Answer: Thank you for your help. We have changed that. Now it reads “Buckwheat treated with 0.1% chitosan for 72 h showed higher levels of phenolic compounds than control samples. Similarly, the germinated buckwheat treated with JA for 48 and 72h produced higher amounts of phenolic compounds than the control samples.”
Introduction:
- Lines 68-70: Reformulate the sentence. Do you mean that the increase in the phenylpropanoids was confirmed by HPLC.
Answer: Thank you for your help. We have rephrased this sentence. Now it reads “In particular, a rapid increase in the concentration of phenolic compounds was observed in JA-elicited cells compared to the control cells after 4 days of JA elicitation [22].”
- Line 71: Sentence "...... compounds compared to the control cells." Citing literature is missing.
Answer: Thank you for your help. We have added Citing literature
Materials and Methods:
- Line 86: were obtained ….from.
Answer: Thank you for your help. We have corrected that. Now it reads “Seeds of common buckwheat were obtained from Jeju Buckwheat Farmers Association Corp. (Je-ju do, Korea).”
- Line 90-91: Report more in detail the conditions of germination (i.e. dark, light etc)
Answer: Thank you for your help. We have corrected that. Now it reads “Two hundred seeds (approximately 4 g) were placed on a filter paper (Whatman, 150-mm diameter) in a petri-dish (AccuResearch Korea, Seoul, South Korea, 150-mm diameter) and then treated with 200 mL of salicylic acid at concentrations of 50, 100, and 150 mg/L, jasmonic acid at concentrations of 50, 100, and 150 μM, and chitosan at concentrations of 0.01, 0.1, and 0.5 % (Figure 1). After incubation under dark condition at 25 °C for 72 h, the germinated buckwheat seeds were harvested and frozen in liquid nitrogen (−196 °C).
- Section 2.2. You should report the phenolics analysed. Moreover, there is needed to report in detail the standards used. Since you do quantitative analysis, the method used should be validated for the substrate analysed in the present study. If you perform it you should report the data.
Answer: Thank you for your help. We have added more information on that. Now it reads “The phenylpropanoid contents were identified based on the retention time and spike test, followed by a calculation using respective calibration curves. The linear equations were y = 7.525240872x - 37.38700057 (R² = 0.999702659) for benzoic acid, y = 39.98286487x - 65.70752695 (R² = 0.999891287) for caffeic acid, y = 7.889742787x - 40.24235366 (R² = 0.999881991) for catechin, y = 8.59893686x - 8.335554043 (R² = 0.99999958) for epi-catechin, y = 32.89591693x - 26.17370908 (R² = 0.9999685) for gallic acid, and y = 8.09714215x - 105.546569 (R² = 0.999542863) for Rutin. The results were presented as microgram per milligram dry weight (μg/mg [dw]) with means ± standard deviation of triplicate experiments.”
- Line 98: The temperature used 60°C could affect negatively the amount of phenolics. Why did you choose this temperature? Did you check that extraction in this temperature is safe?
Answer: Thank you for your help. We are really sorry to have a mistake on the temperature. It was 37 °C. The extraction was performed at 37 °C for 60 min according to the method described by Park et al. [1].
- Line 103: In the abstract was mentioned that was monitored the increase in the content of "phenolic compounds" whereas here it was showed that phenylpropanoid content was identified. Rephrase the sentence.
Answer: Thank you for your help. We have changed it. Now it reads “The phenolic compounds were identified based on the retention time and spike test, followed by a calculation using respective calibration curves.”
- Section 2.3. There are not mentioned the groups that are compared but only the test.
Answer: Thank you for your help. We deleted “The analysis of variance (ANOVA)”.
Results:
- Section 3.1: The results are not explained properly. There is not reported first the effect of elicitors on the total concentration of phenolics and then in each phenolic compound. This could bring to the reader more information and show differences among the phenolics studied.
Answer: Thank you for your help. We have added more information. Now it reads “Table 1 shows the effect of elicitors used during germination on the accumulation of phenolic compounds in buckwheat sprouts. As it can be observed, 6 phenolic compounds and 1 organic acid were detected in germinated buckwheat. The treatment with 0.1% chitosan increased the total phenolic content compared with the control, 0.01 and 0.5% chitosan treatments (p < 0.05). In particular, the total phenolics of the germinated buckwheat treated with 0.1% chitosan were approximately 1.23 times higher than the control buckwheat samples (Table 1). In addition, the concentration of gallic acid, catechin, chlorogenic acid, and (-)-epicatechin in the germinated buckwheat treated with 0.1% chitosan were approximately 1.15, 1.72, 1.69, 2.22 times higher than the control, 0.01%, and 0.5% chitosan treatments. The six phenolic compounds were also detected by HPLC in buckwheat treated by JA. The germinated buckwheat treated with JA at the specific concentrations of 50, 100, and 150 µM increased the accumulation of total phenolic compounds. The germinated buckwheat grown in 150 µM of JA showed the highest amount of total phenolics which was approximately 2.47 times higher than that of control. Particularly, the accumulation of gallic acid, rutin, catechin, chlorogenic acid, and (-)-epicatechin were approximately 2.00, 2.38, 1.76, 2.81, and 7.95 times higher in JA-treated buckwheat than in the control buckwheat samples (Table 1). However, SA treatment did not influence the production of phenolic compounds.”
- Line 116: There are reported four numbers for concentration increase and only three concentration (control, 0.01%, 0.05%).
Answer: Thank you for your help. We are sorry to confuse you. This is comparison of the concentration of gallic acid, catechin, chlorogenic acid, and (-)-epicatechin. Therefore, there are four values. Could we keep this sentence in the manuscript ? We think that we need this sentence. If you really want me to change it we are willing to do it. Thank you.
- Table1: Usually letters used to show differences among the means are assigned from the lowest to the highest value. (a assigned to the lowest value and c to the highest among three values). Report properly the letters. Moreover, the same column begins with the control and finishes with SA150mg/L. You have to report that you compare groups of the same treatment within the same column.
Answer: Thank you for your help. We have changed these parts as you commented.
- Section 3.2: Change the title of the section in order to report the time effect. In the form reported here is not showing the progress during germination time.
Answer: Thank you for your help. We have changed these parts as you commented. Now it reads “3.2. Time-course effects of 0.1% chitosan treatment on phenolic compounds of germinated buckwheat” and “Time-course effects of 150 µM jasmonic acid treatment on phenolic compounds of germinated buckwheat”.
- Line 132-134: You report that "As a result, the concentration of total phenolic compounds increased after 72 h and it was the highest accumulation of phenolics compared with the control treatment. " Comparison was made with control 72h? You should clarify it. You should report the results of this statistical analysis on the table 2. That is comparisons between the control and the 0.1% chitosan by using numbers as superscript. The same is valid and for the table 3
Answer: Thank you for your help. We have changed the sentence a little. Now it reads “As a result, the concentration of total phenolic compounds increased after 72 h and it was the highest accumulation of phenolics compared with the control groups.” Furthermore, we think that Duncan multiple test is enough to show significant difference between experimental and control groups according to the time course effect. We have already described “Means with different letters in the same column differ significantly (p < 0.05, Duncan Multiple Range Test (DMRT)” using superscript. Thank you for your valuable comment.
Discussion
- Discussion is merely a literature review and does not discuss the findings reported in the section Results. Moreover, it has repetitions. It reports that according to the literature rutin is the most abundant phenolic compound found in germinated buckwheat but they do not link this fact with the data of the present study. Similarly, the second most abundant compound is epicatechin but there is not mentioned the fact that in the present study there is not found in the second place.
Answer: Thank you for your help. We have rephrased this sentence. Now it reads “Among the detected phenolics in the germinated buckwheat at 72 h after the treatment of 150 µM JA and 0.1 % chitosan, the concentration of rutin, (-)-epicatechin, and chlorogenic acid highly increased. Rutin, the most abundant phenolic compound in the elicited germinated buckwheat, is used as a health supplement and has applications in food industries due to its biological activities, including anti-oxidant, anti-inflammatory, and anti-diabetic activities [28]. Similarly, (-)-epicatechin, the second most abundant compound, has been introduced as a health supplement because it enhances fatigue resistance and oxidative capacity [29,30]. Chlorogenic acid, the third most abundant compound, has been mainly used in food processing and cosmetic industries since the compound possesses anti-carcinogenic, anti-inflammatory, and anti-oxidant functions [31]. Besides, the other identified compounds have been reported to have health beneficial effect, such as anti-cancer and anti-oxidant effects [32-34].” We wanted to describe health beneficial effect of phenolic compounds, identified or increased by the treatment of chitosan and JA, in the germinated buckwheat. Thank you for your valuable comment.
Conclusions:
Here is reported that the aim of the present study is to optimize the concentrations of elicitors and treatment period but this is not showed in the introduction (The aim in the introduction is not the same with the one reported here). Moreover, the authors do not perform any optimization technique such as response surface methodology in both factors studied (elecitors concentration, treatment period) in order to support the fact that they optimized the process. Here is not showed what are the unique findings of the present study.
Answer: Thank you for your valuable comment. We have rephrased this sentence. Now it reads “This study confirmed that JA and chitosan play an important role in the production of phenolic compounds in germinated buckwheat. A total of six phenolics (gallic acid, catechin, chlorogenic acid, caffeic acid, (-)-epicatechin, and rutin) and one organic acid (benzoic acid) were detected in germinated buckwheat. JA and chitosan treatment enhanced the accumulation of phenolic compounds in the germinated buckwheat. Particularly, the treatments with 0.1 % chitosan and 150 µM JA were the most effective on the accumulation of phenolic compounds. According to the time-course analysis, a 72 h chitosan treatment enhanced the production of phenolics. Similarly, the germinated buckwheat treated for 48 and 72h showed higher accumulation of phenolic compounds than control buckwheat. Thus, these results might help build sturdy strategies to enhance the production of phenolics in germinated buckwheat as a good nutritional source for human consumption.”

Reviewer 2 Report
This manuscript deals with the evaluation of the influence of elicitation by jasmonic acid, chitosan and salicylic acid applied during buckwheat germination on the accumulation of phenolic compounds in buckwheat sprouts. The topic of the research is interesting for the Food Science and Technology field and the results obtained can be useful for food industry. The following points should be considered when making the revision:
Title
-The title should be replaced by “Influence of jasmonic acid, chitosan and salicylic acid on accumulation of phenolic compounds in germinated buckwheat (Fagopyrum esculentum Moench”, or by a similar title.
Abstract
-Line 7. Replace the sentence “The present study investigated the effects of elicitors,….”” by “The present study investigated the effects of jasmonic acid (JA), chitosan….”.
-Line 22. Replace the sentence “A treated with 150 μM JA resulted in the accumulation of…” by “The treatment with 150 μM JA enhanced the levels of phenolics in buckwheat sprouts as compared with those observed in control, 50 and 100 μM JA-treated sprouts”.
-Line 27. Replace “In the buckwheat treated with 0.1% chitosan, the levels of phenolic compounds were higher” by “Buckwheat treated with 0.1% chitosan showed higher levels of phenolic compounds than control samples”.
Key words
-Replace “Phenolic compound” by “Phenolic compounds” and add two more key words: “jasmonic acid and chitosan”
Introduction
-Page 1, line 37. Change “and used” by “and consumed”.
-Page 1, line 44. Replace “, and intake and use of” by “and the use of these constituents as functional ingredients have been….”.
-Page 2, line 45. Replace “has” by “have”.
-Page 2, lines 57-59. Replace the sentence “Furthermore, these secondary metabolites…daily human diet” by “These secondary metabolites are well-known components of other food sources used in the daily human diet”.
-Page 2, line 59. Replace “It exhibits” by “They exhibit”.
-Page 2, line 63. Change “…promoted by the elicitations of chitosan…” by ““…promoted by the elicitations by chitosan…”.
-Page 2, lines 68-70. Please, rewrite the sentence “In particular, phenylpropanoids using HPLC….compared to the control cells”. The sentence has no sense.
-Page 2, lines 79-80. Remove “using high-performance liquid chromatography (HPLC).
Materials and Methods
-Page 3, line 86. The sentence “Seeds of common buckwheat were obtained…” is not clear. What does it mean JEJU buckwheat? , is it the type of buckwheat?, Is it the company that provides the buckwheat seeds?. Please rewrite the sentence for improving the understanding.
-Page 3, lines 91-92. Include the temperature used during germination process.
-Page 3, line 96. Replace “Park et al. (2017) [1]” by “Park et al. [1]”.
-Page 3, line 102. Replace “were used as described by Park et al. 2017” by “were performed as described by Park et al. [1]”.
-Page 3, line 107. Remove “evaluation” after “ANOVA”.
Results
-Page 3, lines 111-113. Replace “In this study the germinated buckwheat …were detected by HPLC analysis” by “Figure 1 shows the effect of elicitors used during germination on the accumulation of phenolic compounds in buckwheat sprouts. As it can be observed, 7 phenolic compounds were detected in germinated buckwheat”.
-Page 3, line 114. Remove “Furthermore”.
-Page 3, lines 119-120. Change the sentence “When treated with jasmonic acid, all seven….” By “The seven phenolic compounds were also detected by HPLC in buckwheat treated by JA”.
-Page 4, lines 131-132. Replace the sentence “in 0.1% chitosan treatment” by “after 0.1% chitosan treatment” and “this study was progressed by a time course experiment….72 h” by “the accumulation of phenolic compounds in 0.1% chitosan-treated buckwheat was studied throughout the germination process (3, 12, 24, 48 and 72 h).”
-Page 4, line 136. The sentence “than in buckwheat under control and other treatment conditions” is not true since other treatment conditions caused higher accumulation of phenolic compounds than the treatment by chitosan for 72 h”. I suggest to remove “and other treatment conditions” from the sentence.
-Page 5, lines 144-145. Replace the sentence “The time course experiments at 6, 12, 24, 48, 72 h were conducted…150 μM jasmonic acid” by “The time course experiments at 6, 12, 24, 48 and 72 h were also conducted in buckwheat germinated in the presence of 150 μM jasmonic acid”.
-Page 5, line 146. Replace “was increased” by “increased”. Authors stated that accumulation of phenolic compounds increased after 48 h, but, actually, it increased after 72 h.
Discussion
-Page 6, lines 158-161. Replace “This analysis result was consistent with the previous studies…in common buckwheat flours” by “These results are consistent with previous studies reporting the identification of gallic acid, chlorogenic acid, catechin, caffeic acid, (-)-epicatechin, and rutin in common buckwheat sprouts [22] and flours [23].”
-Page 6, line 162. Remove “The” from the sentence “..in the buckwheat honeys”.
-Page 6, line 167. Replace “pharmacological” by “biological”.
-Page 6, line 178. Remove “elictiors” before “trigger”.
-Page 6, line 181. Remove “the comma” after “including”.
Conclusions
-Page 7, line 201. Change “of elicitors” by “by elicitors”.
-Page 7, line 204. Remove “Apparently”.
-Page 7, line 208. Change “…the germinated buckwheat treated with 48 and 72h showed the accumulation of higher levels of phenolic compounds” by “…the germinated buckwheat treated for 48 and 72h showed higher accumulation of phenolic compounds than control buckwheat”.
-Page 7, lines 212-214. Please, rewrite the sentence “Therefore, ….bound form”. It has no sense.
Figure 1. Replace the legend by “Buckwheat germinated for 72 hours”.
Author Response
This manuscript deals with the evaluation of the influence of elicitation by jasmonic acid, chitosan and salicylic acid applied during buckwheat germination on the accumulation of phenolic compounds in buckwheat sprouts. The topic of the research is interesting for the Food Science and Technology field and the results obtained can be useful for food industry. The following points should be considered when making the revision:
Answer: Thank you for your valuable comment. We are honored to receive your feedback and we are sure that your comment will significantly improve the paper.
Title
-The title should be replaced by “Influence of jasmonic acid, chitosan and salicylic acid on accumulation of phenolic compounds in germinated buckwheat (Fagopyrum esculentum Moench”, or by a similar title.
Answer: Thank you for your help. We have changed the title as you commented.
Abstract
-Line 7. Replace the sentence “The present study investigated the effects of elicitors,….”” by “The present study investigated the effects of jasmonic acid (JA), chitosan….”.
Answer: Thank you for your help. We have changed this sentence as you commented.
-Line 22. Replace the sentence “A treated with 150 μM JA resulted in the accumulation of…” by “The treatment with 150 μM JA enhanced the levels of phenolics in buckwheat sprouts as compared with those observed in control, 50 and 100 μM JA-treated sprouts”.
Answer: Thank you for your help. We have changed this sentence as you commented.
-Line 27. Replace “In the buckwheat treated with 0.1% chitosan, the levels of phenolic compounds were higher” by “Buckwheat treated with 0.1% chitosan showed higher levels of phenolic compounds than control samples”.
Answer: Thank you for your help. We have changed this sentence as you commented.
Key words
-Replace “Phenolic compound” by “Phenolic compounds” and add two more key words: “jasmonic acid and chitosan”
Answer: Thank you for your help. We have added more key words as you commented.
Introduction
-Page 1, line 37. Change “and used” by “and consumed”.
Answer: Thank you for your help. We have changed that as you commented.
-Page 1, line 44. Replace “, and intake and use of” by “and the use of these constituents as functional ingredients have been….”.
Answer: Thank you for your help. We have changed this sentence as you commented.
-Page 2, line 45. Replace “has” by “have”.
Answer: Thank you for your help. We have corrected this part as you commented.
-Page 2, lines 57-59. Replace the sentence “Furthermore, these secondary metabolites…daily human diet” by “These secondary metabolites are well-known components of other food sources used in the daily human diet”.
Answer: Thank you for your help. We have changed this sentence as you commented.
-Page 2, line 59. Replace “It exhibits” by “They exhibit”.
Answer: Thank you for your help. We have changed that as you commented.
-Page 2, line 63. Change “…promoted by the elicitations of chitosan…” by ““…promoted by the elicitations by chitosan…”.
Answer: Thank you for your help. We have changed that as you commented.
-Page 2, lines 68-70. Please, rewrite the sentence “In particular, phenylpropanoids using HPLC….compared to the control cells”. The sentence has no sense.
Answer: Thank you for your help. We have rephrased this sentence. Now it reads “In particular, a rapid increase in the concentration of phenolic compounds was observed in JA-elicited cells compared to the control cells after 4 days of JA elicitation [22].”
-Page 2, lines 79-80. Remove “using high-performance liquid chromatography (HPLC).
Answer: Thank you for your help. We have removed that as you commented.
Materials and Methods
-Page 3, line 86. The sentence “Seeds of common buckwheat were obtained…” is not clear. What does it mean JEJU buckwheat? , is it the type of buckwheat?, Is it the company that provides the buckwheat seeds?. Please rewrite the sentence for improving the understanding.
Answer: Thank you for your help. We are sorry to confuse you. This is a company. We have added more information on that this sentence as you commented. Now it reads “Seeds of common buckwheat were obtained from Jeju Buckwheat Farmers Association Corp. (Je-ju do, Korea).”.
-Page 3, lines 91-92. Include the temperature used during germination process.
Answer: Thank you for your help. We have added more information on that this sentence as you commented. Now it reads “After incubation at 25 °C for 72 h, the germinated buckwheat seeds were harvested and frozen in liquid nitrogen (−196 °C).”
-Page 3, line 96. Replace “Park et al. (2017) [1]” by “Park et al. [1]”.
Answer: Thank you for your help. We have changed that as you commented.
-Page 3, line 102. Replace “were used as described by Park et al. 2017” by “were performed as described by Park et al. [1]”.
Answer: Thank you for your help. We have changed that as you commented.
-Page 3, line 107. Remove “evaluation” after “ANOVA”.
Answer: Thank you for your help. We have removed that as you commented.
Results
-Page 3, lines 111-113. Replace “In this study the germinated buckwheat …were detected by HPLC analysis” by “Figure 1 shows the effect of elicitors used during germination on the accumulation of phenolic compounds in buckwheat sprouts. As it can be observed, 7 phenolic compounds were detected in germinated buckwheat”.
Answer: Thank you for your help. We have changed that as you commented.
-Page 3, line 114. Remove “Furthermore”.
Answer: Thank you for your help. We have removed that as you commented.
-Page 3, lines 119-120. Change the sentence “When treated with jasmonic acid, all seven….” By “The seven phenolic compounds were also detected by HPLC in buckwheat treated by JA”.
Answer: Thank you for your help. We have changed that as you commented.
-Page 4, lines 131-132. Replace the sentence “in 0.1% chitosan treatment” by “after 0.1% chitosan treatment” and “this study was progressed by a time course experiment….72 h” by “the accumulation of phenolic compounds in 0.1% chitosan-treated buckwheat was studied throughout the germination process (3, 12, 24, 48 and 72 h).”
Answer: Thank you for your help. We have changed those parts as you commented.
-Page 4, line 136. The sentence “than in buckwheat under control and other treatment conditions” is not true since other treatment conditions caused higher accumulation of phenolic compounds than the treatment by chitosan for 72 h”. I suggest to remove “and other treatment conditions” from the sentence.
Answer: Thank you for your help. We have removed that as you commented.
-Page 5, lines 144-145. Replace the sentence “The time course experiments at 6, 12, 24, 48, 72 h were conducted…150 μM jasmonic acid” by “The time course experiments at 6, 12, 24, 48 and 72 h were also conducted in buckwheat germinated in the presence of 150 μM jasmonic acid”.
Answer: Thank you for your help. We have changed that as you commented.
-Page 5, line 146. Replace “was increased” by “increased”. Authors stated that accumulation of phenolic compounds increased after 48 h, but, actually, it increased after 72 h.
Answer: Thank you for your help. We have changed these parts as you commented. Now it reads “As a result, the accumulation of total phenolic compounds increased after 72 h.”.
Discussion
-Page 6, lines 158-161. Replace “This analysis result was consistent with the previous studies…in common buckwheat flours” by “These results are consistent with previous studies reporting the identification of gallic acid, chlorogenic acid, catechin, caffeic acid, (-)-epicatechin, and rutin in common buckwheat sprouts [22] and flours [23].”
Answer: Thank you for your help. We have changed these parts as you commented. Now it reads “As a result, the accumulation of total phenolic compounds increased after 72 h.”.
-Page 6, line 162. Remove “The” from the sentence “..in the buckwheat honeys”.
Answer: Thank you for your help. We have removed that as you commented.
-Page 6, line 167. Replace “pharmacological” by “biological”.
Answer: Thank you for your help. We have changed that as you commented.
-Page 6, line 178. Remove “elictiors” before “trigger”.
Answer: Thank you for your help. We have removed that as you commented.
-Page 6, line 181. Remove “the comma” after “including”.
Answer: Thank you for your help. We have removed that as you commented.
Conclusions
-Page 7, line 201. Change “of elicitors” by “by elicitors”.
Answer: Thank you for your help. We have changed that as you commented.
-Page 7, line 204. Remove “Apparently”.
Answer: Thank you for your help. We have removed that as you commented.
-Page 7, line 208. Change “…the germinated buckwheat treated with 48 and 72h showed the accumulation of higher levels of phenolic compounds” by “…the germinated buckwheat treated for 48 and 72h showed higher accumulation of phenolic compounds than control buckwheat”.
Answer: Thank you for your help. We have changed that as you commented.
-Page 7, lines 212-214. Please, rewrite the sentence “Therefore, ….bound form”. It has no sense.
Answer: Thank you for your help. We think that these sentences are not really related to this study. That’s why we deleted these sentences. Thank you.
Figure 1. Replace the legend by “Buckwheat germinated for 72 hours”.
Answer: Thank you for your help. We have changed that as you commented.

Reviewer 3 Report
comments and suggestions are in the uploaded files. The English is pretty good but a native English speaker could improve the article quite a bit.
It would be wise to include the structures of benzoic acid, gallic acid, caffeic acid, chlorogenic acid, catechin, epicatechin, and rutin somewhere.

Author Response
comments and suggestions are in the uploaded files. The English is pretty good but a native English speaker could improve the article quite a bit.
Answer: Thank you for your valuable comment. We are honored to receive your feedback and we are sure that your comment will significantly improve the paper.
It would be wise to include the structures of benzoic acid, gallic acid, caffeic acid, chlorogenic acid, catechin, epicatechin, and rutin somewhere.
Answer: Thank you for your valuable comment. We have attached a file containing chemical structures of phenolic compounds and benzoic acid.
page | line(s) | from | to | comments |
1 | 36 | Fagopyrum esculentum Moench (common buckwheat) | Add family (Polygonaceae) here or under method
Answer: Thank you for your comment. We have added in as you commented. Now it reads “Fagopyrum esculentum Moench (common buckwheat), belonging to the Polygonaceae family, is an important pseudocereal cultivated and consumed in East Asian countries.” | |
1 | 39 | ref 4 | is not about cytoprotective, anti-thrombotic, and anti-carcinogenic activities
Answer: Thank you for your comment. We have added three more references.
Potapovich, A. I., & Kostyuk, V. A. (2003). Comparative study of antioxidant properties and cytoprotective activity of flavonoids. Biochemistry (Moscow), 68(5), 514-519.
Choi, Jun-Hui, Dae-Won Kim, Se-Eun Park, Hyo-Jeong Lee, Ki-Man Kim, Kyung-Je Kim, Myung-Kon Kim, Sung-Jun Kim, and Seung Kim. "Anti-thrombotic effect of rutin isolated from Dendropanax morbifera Leveille." Journal of bioscience and bioengineering 120, no. 2 (2015): 181-186. Deschner EE, Ruperto J, Wong G, Newmark HL. Quercetin and rutin as inhibitors of azoxymethanol-induced colonic neoplasia. Carcinogenesis 1991;12:1193–6
| |
2 | 55 | phenylpropanoid pathway contributes | The phenylpropanoid pathway is the only source of flavonoids (and several other classes)
Answer: Thank you for your comment. We have changed the sentence. Now it reads “They are derived from the phenylpropanoid pathway”. | |
2 | 61 | Abiotic stresses such as signal molecules or elicitors | Activated by abiotic stresses, signal molecules or elicitors
Answer: Thank you for your comment. We have changed it as you commented. | |
2 | 67 | Hypericum perforatum L. | Hypericum perforatum L. (St. John's wort)
Answer: Thank you for your comment. We have changed it as you commented. | |
2 | 73-74 | tomato seedlings as well as the anthocyanin production in hypocotyls and fruits | References 19-21 make no mention of tomato seedlings
Answer: Thank you for your comment. We have added one more reference.
Guo J and Wang MH, Ultraviolet A-specific induction of anthocyanin biosynthesis and PAL expression in tomato (Solanum lycopersicum L.). Plant Growth Regul 62:1–8 (2010) | |
2 | 75 | exposed to ultraviolet-B (UV-B) radiation | not mentioned in reference 20?
Answer: Thank you for your comment. No. It is not mentioned in ref 20. | |
3-4 | Table 1 | list compounds alphabetically or by increasing complexity (benzoic acid, gallic acid, caffeic acid, chlorogenic acid, catechin, epicatechin, rutin)
Answer: Thank you for your comment. We have listed these compounds alphabetically. | ||
4-5 | Table 2 | list compounds alphabetically or by increasing complexity (benzoic acid, gallic acid, caffeic acid, chlorogenic acid, catechin, epicatechin, rutin)
Answer: Thank you for your comment. We have listed these compounds alphabetically. | ||
5-6 | Table 3 | list compounds alphabetically or by increasing complexity (benzoic acid, gallic acid, caffeic acid, chlorogenic acid, catechin, epicatechin, rutin)
Answer: Thank you for your comment. We have listed these compounds alphabetically. | ||
6 | 158 | benzoic acid | is not a true phenolic compound
Answer: Thank you for your comment. We have described information on that. Now it reads “six phenolic compounds (gallic acid, catechin, chlorogenic acid, caffeic acid, (-)-epicatachin, and rutin) and one organic acid (benzoic acid) were detected in germinated buckwheat.” | |
7 | 208 | the germinated buckwheat treated with 48 and 72h | 48 and 72h germinated buckwheat
Answer: Thank you for your comment. The another reviewer wanted me to replace this sentence by “the germinated buckwheat treated for 48 and 72h”. If you want to change the sentence as you commented, I am willing to change it. Thank you so much. |
Round 2
Reviewer 1 Report
Manuscript foods-452364 “Influence of Chitosan and Jasmonic Acid on Phenylpropanoid Accumulation in Germinated Buckwheat (Fagopyrum esculentum Moench)”.
As reported previously the manuscript lacks novelty since are not used novel elicitors as well as the results obtained did not show something new that is not already reported in the literature.
The changes made did not improved the quality of the manuscript. It is still confusing. There are still issues related to the validation of the HPLC method. Moreover, it seems that authors are confusing about the statistical analysis performed. Duncan test is a post hock test performed after ANOVA analysis because ANOVA reports if there is a difference in means without giving information which means are different. Duncan’s Multiple Range test measures specific differences between pairs of means. Moreover, the authors report optimization but they do not perform any optimization technique such as response surface methodology in order to support the fact that they optimized the process. This fact represents a deficiency that affects the quality of the manuscript.
The discussion is still not convincing.
Other concerns to be addressed:
Abstract: Is reporting the findings of the work without giving to the reader a clean view of the treatment performed and the main achievement of the present work.
Materials and Methods: Section 2.2. There is needed to report the standards used (company, city, country), concentrations used to prepare the calibrations curves as well as recoveries for each phenolic compound. It was not mentioned how was performed the validation of the method. Moreover, the equations reported for the calibration curves should be rounded to the 4 decimals. It is useless to report 8 decimals.
Section 2.3. Why did you remove anova. The Duncan’s Multiple Range Test is a post hoc test performed after the ANOVA test. Moreover, you must report the groups of means that you are comparing.
Results:
The response given to the question raised to in the Line 116 is not satisfactory. The sentence in the form given is confusing.
The statistical analysis is not reported properly. There is still not reported that are compared groups of the same treatment within the same column.
The answer given to the question Line 132-134 is not satisfactory. There are different controls (6, 12, 24, 48 and 72h). How was done the comparison. It is not clear.
Discussion
- Discussion is lacking to explain the finding and is still a literature review
Author Response
As reported previously the manuscript lacks novelty since are not used novel elicitors as well as the results obtained did not show something new that is not already reported in the literature.
The changes made did not improved the quality of the manuscript. It is still confusing. There are still issues related to the validation of the HPLC method.
Answer: Thank you for your comment. We have added more information on the HPLC method below and in the manuscript.
Moreover, it seems that authors are confusing about the statistical analysis performed. Duncan test is a post hock test performed after ANOVA analysis because ANOVA reports if there is a difference in means without giving information which means are different. Duncan’s Multiple Range test measures specific differences between pairs of means.
Answer: We are really sorry that we did not really understand what you said (Section 2.3. There are not mentioned the groups that are compared but only the test). We have tried to understand your comment. However, we could not understand it. We are really sorry for that. Therefore, we have returned it to the original.
Moreover, the authors report optimization but they do not perform any optimization technique such as response surface methodology in order to support the fact that they optimized the process. This fact represents a deficiency that affects the quality of the manuscript.
Answer: Thank you for your comment. We think that it is enough to describe the influence of chitosan, salicylic acid, and jasmonic acid on phenylpropanoid accumulation in germinated buckwheat (Fagopyrum esculentum Moench) as well as provide optimal concentration of chitosan, salicylic acid, and jasmonic acid for the production of phenolics in the buckwheats. Previously, many studies [1-8] successfully reported that the optimal concentrations of elicitors for the production of various secondary metabolites without optimization technique such as response surface methodology. Therefore, we think that it is not necessary to perform the technique.
1. Świeca, M. (2015). Production of ready-to-eat lentil sprouts with improved antioxidant capacity: optimization of elicitation conditions with hydrogen peroxide. Food chemistry, 180, 219-226.
2. Baenas, N., García-Viguera, C., & Moreno, D. A. (2014). Biotic elicitors effectively increase the glucosinolates content in Brassicaceae sprouts. Journal of agricultural and food chemistry, 62(8), 1881-1889.
3. Baenas, N., Villaño, D., García-Viguera, C., & Moreno, D. A. (2016). Optimizing elicitation and seed priming to enrich broccoli and radish sprouts in glucosinolates. Food chemistry, 204, 314-319.
4. im, H. J., Park, K. J., & Lim, J. H. (2011). Metabolomic analysis of phenolic compounds in buckwheat (Fagopyrum esculentum M.) sprouts treated with methyl jasmonate. Journal of agricultural and food chemistry, 59(10), 5707-5713
5. Randhir, R., Lin, Y. T., & Shetty, K. (2004). Phenolics, their antioxidant and antimicrobial activity in dark germinated fenugreek sprouts in response to peptide and phytochemical elicitors. Asia Pacific journal of clinical nutrition, 13(3).
6. Zhao, J. L., Zou, L., Zhong, L. Y., Peng, L. X., Ying, P. L., Tan, M. L., & Zhao, G. (2015). Effects of polysaccharide elicitors from endophytic Bionectria pityrodes Fat6 on the growth and flavonoid production in tartary buckwheat sprout cultures. Cereal research communications, 43(4), 661-671.
7. Chen, H., & Chen, F. (2000). Effect of yeast elicitor on the secondary metabolism of Ti-transformed Salvia miltiorrhiza cell suSahu, R., Gangopadhyay, M., & Dewanjee, S. (2013). Elicitor-induced rosmarinic acid accumulation and secondary metabolism enzyme activities in Solenostemon scutellarioides. Acta physiologiae plantarum, 35(5), 1473-1481. spension cultures. Plant Cell Reports, 19(7), 710-717.
8. Randhir, R., Kwon, Y. I., & Shetty, K. (2009). Improved health-relevant functionality in dark germinated Mucuna pruriens sprouts by elicitation with peptide and phytochemical elicitors. Bioresource technology, 100(19), 4507-4514.
9. Al-Dhabi, N. A., Arasu, M. V., Kim, S. J., RomijUddin, M., Park, W. T., Lee, S. Y., & Park, S. U. (2015). Methyl jasmonate-and light-induced glucosinolate and anthocyanin biosynthesis in radish seedlings. Natural product communications, 10(7), 1934578X1501000719.
The discussion is still not convincing.
Other concerns to be addressed:
Abstract: Is reporting the findings of the work without giving to the reader a clean view of the treatment performed and the main achievement of the present work.
Answer: Thank you for your comment. We have added more information on that. Now it reads “The present study investigated the effects of jasmonic acid (JA), chitosan, and salicylic acid (SA), on the accumulation of phenolic compounds in germinated buckwheat. A total of six phenolics were detected in the buckwheat treated with different concentrations of SA (50, 100, and 150 mg/L), JA (50, 100, and 150 μM), and chitosan (0.01, 0.1, and 0.5 %), using high-performance liquid chromatography (HPLC). The treatment with 0.1 % chitosan resulted in the accumulation of the highest levels of phenolic compounds as compared with the control- and 0.01 and 0.5 % chitosan treatments. The treatment with 150 μM JA enhanced the levels of phenolics in buckwheat sprouts as compared with those observed in control, 50 and 100 μM JA-treated sprouts. However, the SA treatment did not affect the production of phenolic compounds. After optimizing the treatment concentrations of elicitors (chitosan and JA), a time-course analysis of the phenolic compounds detected in the germinated buckwheat treated with 0.1 % chitosan and 150 μM JA was performed. Buckwheat treated with 0.1% chitosan for 72 h showed higher levels of phenolic compounds than all control samples. Similarly, the germinated buckwheat treated with JA for 48 and 72h produced higher amounts of phenolic compounds than all control samples. This study is to elucidate the influence of SA, JA, and chitosan on the production of phenolic compounds and suggests that the treatment with optimal concentrations of chitosan and JA for an optimal time period improved the production of phenolic compounds in germinated buckwheat.”.
Materials and Methods: Section 2.2. There is needed to report the standards used (company, city, country), concentrations used to prepare the calibrations curves as well as recoveries for each phenolic compound. It was not mentioned how was performed the validation of the method. Moreover, the equations reported for the calibration curves should be rounded to the 4 decimals. It is useless to report 8 decimals.
Answer: Thank you for your comment. We are sorry for the mistake. We have added more information on that. Now it reads “The linear equations were y = 7.5252x - 37.3870 (R² = 0.9997, recovery value = 102.81 ± 5.32 %) for benzoic acid, y = 39.9829x - 65.7075 (R² = 0.9999, recovery value = 102.14 ± 3.67 %) for caffeic acid, y = 7.8897x - 40.2424 (R² = 0.9999, recovery value = 104.09 ± 11.25 %) for catechin, y = 8.5989x - 8.3356 (R² = 0.9999, recovery value = 100.28 ± 0.80 %) for epi-catechin, y = 32.8959x - 26.1737 (R² = 0.9999, recovery value = 96.57 ± 2.51 %) for gallic acid, and y = 8.0971x - 105.5466 (R² = 0.9995, recovery value = 104.61 ± 11.17 %) for rutin. The external standards were purchased from Sigma-Aldrich Co., Ltd. (St. Louis, MO, USA).”.
Section 2.3. Why did you remove anova. The Duncan’s Multiple Range Test is a post hoc test performed after the ANOVA test. Moreover, you must report the groups of means that you are comparing.
Answer: Thanks for your comments. We are really sorry that we did not really understand what you said (Section 2.3. There are not mentioned the groups that are compared but only the test). We have tried to understand your comment. However, we could not understand it. We are really sorry for that. Therefore, we have returned it to the original. If you meant that we should report means of the groups, the results were presented as microgram per milligram dry weight (μg/mg [dw]) with means ± standard deviation of triplicate experiments.
Results:
The response given to the question raised to in the Line 116 is not satisfactory. The sentence in the form given is confusing.
Answer: Thanks for your comments. We agree with your opinion and have changed the sentence. Now it reads “In addition, the concentration of gallic acid, catechin, chlorogenic acid, and (-)-epicatechin in the germinated buckwheat treated with 0.1% chitosan were approximately 15.86, 1.72, 1.64, 2.17 times higher than those of the control.”.
The statistical analysis is not reported properly. There is still not reported that are compared groups of the same treatment within the same column.
Answer: Thank you for your comment. We have already provided the data in the Table 1 of the first revised manuscript. Could you please check carefully Table 1?
The answer given to the question Line 132-134 is not satisfactory. There are different controls (6, 12, 24, 48 and 72h). How was done the comparison. It is not clear.
Answer: Thank you for your comment. We have added a little more information on that. Now it reads “As a result, the concentration of total phenolic compounds increased after 72 h and it was the highest accumulation of phenolics compared with all the control groups (6, 12, 24, 48 and 72h).”
Discussion
- Discussion is lacking to explain the finding and is still a literature review
Answer: Thank you for your valuable comment. We have added more information on that. We think that this discussion is enough to interpret our findings since the other two referees were satisfied with it and that this is not a literature review since DISCUSSION contains findings by comparing with the findings in prior studies. We hope that you will consider it favorably. Thank you so much again.
Now it reads" This time course analysis revealed that chitosan and JA gradually enhanced production of phenolic compounds in the germinated buckwheat. We carefully suggested that it might be due to increased gene expression levels of phenylpropanoid-related genes by the chitosan and JA elicitation since our previous studies reported that the methyl jasmonate increased gene expression levels of phenlypropanoid-related genes and enhanced the accumulation of phenolic compounds in radish sprouts [27] and in Agastache rugosa Kuntze [28], respectively. Furthermore, Chen et al. (2009) reported the increased expression of phenylpropanoid and flavonoid biosynthesis genes and in soybean sprouts treated with chitosan [29]. Among the detected phenolics in the germinated buckwheat at 72 h after the treatment of 150 µM JA and 0.1 % chitosan, the concentration of rutin, (-)-epicatechin, and chlorogenic acid significantly increased. Rutin, the most abundant phenolic compound in the elicited germinated buckwheat, is used as a health supplement and has applications in food industries due to its biological activities, including anti-oxidant, anti-inflammatory, and anti-diabetic activities [30]. Similarly, (-)-epicatechin, the second most abundant compound, has been introduced as a health supplement because it enhances fatigue resistance and oxidative capacity [31,32]. Chlorogenic acid, the third most abundant compound, has been mainly used in food processing and cosmetic industries since the compound possesses anti-carcinogenic [33], anti-inflammatory [34], and anti-oxidant functions [35]. Besides, the other identified compounds have been reported to have health beneficial effect, such as anti-cancer and anti-oxidant effects [36-38].
Elicitation is considered one of best strategies to stimulate secondary metabolites. The accumulation of secondary metabolites from either parts of parent or transformed plants is greatly dependent on the sources of their origin; however, it might be influenced by the treatments as well as environmental factors. Elicitors when in contact with the cells of higher plants, trigger an increase in the production of pigments, flavones, phytoalexins, and other defense related compounds [39-42]. This study revealed that treatment with elicitors chitosan or jasmonic acid could enhance the production of phenolic compounds in germinated buckwheat. This finding was consistent with previous studies of Park et al. (2017) [1] and Kim et al. (2011) [43], who reported the enhancement of phenolics in the sprouts of common buckwheat by treatment with indoleacetic acid and methyl jasmonic acid, respectively. Li et al. 2015 [44] reported the positive effect of the exogenous application of sucrose on the flavonoid contents of common buckwheat seedlings. Lim et al. 2012 [45] reported that the sodium chloride (NaCl) treatment enhanced both phenylpropanoid and carotenoid production in buckwheat sprouts. In addition, elicitors could stimulate the biosynthesis of phenylpropanoid compounds in tartary buckwheat (F. tataricum (L.) Gaertn.). Zhao et al. 2015 [46] reported increase in flavonoid production in sprout cultures under treatment of polysaccharide elicitors. Sun et al. 2012 [47] reported that the treatment with salicylic acid resulted in an increase in rutin production in tartary buckwheat leaves. Li et al. (2017) [48] also described that exogenous ethephon application enhanced phenylpropanoid biosynthesis. Furthermore, Park et al. (2016) [49] reported that the treatment with auxins improved anthocyanin production in the hairy root cultures of tartary buckwheat.”.

Reviewer 3 Report
Mostly minor changes in attached file. It would be nice to add structures to chitosan and jasmonic acid as it illustrates how different the elicitors are from the products produced.

Author Response
Foods 452364 peer review v2
Answer: Thank you for your valuable comment. We respect your expertise in this research field. We have tried to correct what you commented. We totally agree with your suggestions and opinion. We appreciate your help.
page | line(s) | from | to | comments |
1 | 40 | reference 2 | is not about rutin content, reference 7 would be better
Answer: Thanks your comments. We have checked it. We have changed ‘reference 7’ to ‘reference 2’. | |
2 | 45 | check grammar
Answer: Thanks your comments. We have changed ‘have’ to ‘has’. | ||
2 | 44-54 | lines 45, 50, 52, 53 | please have native English speaker review Answer: Thanks your comments. We have corrected those parts and highlighted the sentences with yellow color.
In line 45, ‘have’ has been changed to ‘has’. In line 50, ‘a role in in-vivo’ has been changed to ‘a role of antioxidant protection’. In line 52-53, ‘Such an continuous~’ has been changed to ‘Such a continuous~’. | |
2 | 59 | of other food | of food | Answer: Thanks your comments. We have corrected it. |
2 | 59 | check grammar
Answer: Thanks your comments. We have corrected those parts and highlighted the sentences with yellow color.
We have changed ‘exhibits’ to ‘exhibit’. | ||
2 | 65 | reference 20 | appears to be only about "The addition of chitosan, chitosan oligosaccharide and alginate oligosaccharide to the culture of P. ginseng hairy roots " Answer: Thanks for your comments. We are really sorry for the mistakes on reference. We have changed the reference with correct one. [Chakraborty, M.; Karun, A.; Mitra, A. Accumulation of phenylpropanoid derivatives in chitosan-induced cell suspension culture of Cocos nucifera. J. Plant Physiol. 2009, 166, 63-71.] | |
2 | 73-75 | reference 24 | is about Lactuca sativa not Hypericum perforatum
Answer: Thanks your comments. We are sorry for the mistake. We have changed the reference with the correct one. | |
2 | 76-79 | To our knowledge, no previous reports have documented the influence of chitosan, salicylic acid, and jasmonic acid on the accumulation of flavonoids in germinated buckwheat. Thus, the current study aimed to elucidate the effect of chitosan, salicylic acid, and jasmonic acid on the production of phenolic compounds in germinated buckwheat | To our knowledge, no previous reports have documented the influence of chitosan, salicylic acid, and jasmonic acid on the accumulation of flavonoids in germinated buckwheat which the current study aims to elucidate. | Redundancy
Answer: Thanks your comments. We have corrected it as you commented. |
3 | 116 | As it can be observed, 6 phenolic | 6 phenolic | no chromatogram or table of retention times is provided Answer: Thanks your comments. We have added information on retention time of the detected compounds in table 1. |
3-5 | results | the tables are labeled accurately but you should put in a disclaimer in the text to the effect that "for the purpose of discussion total phenolics includes benzoic acid"
Answer: Thank you for your comment. We have added more information on that. Now it reads “Even though benzoic acid does not belong to phenolic compound, the total phenolic compound of all samples were described, including benzoic acid.” from the beginning of 3. Results. | ||
6 | 172-173 | highly increased | significantly increased | Answer: Thank you for your comments. We have corrected it. |
6 | 179 | reference 31 | abstract only mentions antioxidant functions Answer: Thanks for your comments. We have added two more references about anti-carcinogenic and anti-inflammatory. Thus, previous ref31 was changed to 32. | |
7 | 187 | elicitors including chitosan | elicitors chitosan | Answer: Thanks for your comments. We have changed it as you commented. |
8 | 240 | 5Kreft, | Kreft, | Answer: Thanks your comments. We have corrected it. |
9 | 307 | Rootsa | Roots | Answer: Thanks your comments. We have corrected it. |
